# Underground Wireless Data Transmission Using 433-MHz LoRa for Agriculture

**DOI:** 10.3390/s19194232

**Published:** 2019-09-29

**Authors:** Marcus Hardie, Donald Hoyle

**Affiliations:** 1Tasmanian Institute of Agriculture, University of Tasmania, Hobart TAS 7000, Australia; 2Cooperative Research Centre for High Performance Soils, Callaghan NSW 2308, Australia; 3School of Agricultural Science, University of Tasmania, Hobart TAS 7000, Australia; donhoyle717@gmail.com

**Keywords:** wireless sensor network, belowground, agritech, agricultural technology

## Abstract

Wireless underground sensor networks (WUSNs) have potential for providing real-time data for agriculture and other industries without exposing sensors and communication infrastructure to damage. However, soil is a difficult environment for radio communication due to its dielectric properties and variable moisture content. Low-power, wide-area network (LPWAN) technologies have been used to develop aboveground sensor networks for many industries, but have not yet been successfully developed for underground applications. In this study, we developed a 433-MHz LoRa-based testbed for evaluating both underground-to-underground (UG2UG) and underground-to-aboveground (UG2AG) wireless communication technologies in four in situ soils. The maximum transmission distance for UG2UG operation was 4–20 m depending on soil type, whilst UG2AG operation was able to communicate up to 100–200 m, depending on the operating variables and soil properties. Signal quality and the maximum transmission distance were influenced by transmitter (TX) burial depth, TX power, data rate, receiver (RX) antenna type, and to a lesser extent, soil parameters. Results suggest that with improvements to power management, the development of 433-MHz LoRa-based UG2AG WUSNs for agricultural applications is readily achievable, whilst UG2UG applications appear unlikely without substantial improvement in transmission distance.

## 1. Introduction

In agriculture, wireless sensor networks (WSNs) have been widely promoted as a means to improve on-farm yield and profitability through the provision of real-time or on-demand sensed data [1,2,3]. WSNs operate by wirelessly sending data from spatially distributed nodes to either a relay node or a gateway. Most WSN applications in agriculture involve sensing soil moisture and/or weather, in which sensed data is usually used for irrigation decision support [1,3]. A range of commercially available WSN solutions exists for agricultural applications; e.g., the Plexus mesh-networking radio system (Measurement Engineering Australia Pty Ltd, Toowoomba, Australia) and Teralytic 26 sensor LoRa node (Teralytic, New York, NY, USA). Despite the commercial availability of soil moisture WSNs, on-farm use of moisture sensor technology, let alone WSNs, is actually quite poor. Only 22% of Australian farmers use any sort of soil moisture monitoring product [4]. Lack of adoption is in large part due to the risk of damage to in-field sensors and WSN nodes by machinery, extreme weather events, grazing animals, and pests [5,6]. One solution is to bury all sensor and communication components below the depth of cultivation to form a wireless underground sensor network (WUSN) [5,7]. WUSNs operate by wirelessly sending data through both the soil and air between two buried sensor nodes as underground-to-underground ground communication (UG2UG), or from a buried node through the ground–air boundary to an aboveground node as underground-to-aboveground communication (UG2AG) [6,7,8]. While aboveground WSNs have been readily commercialized, WUSNs are still at the research stage of development [9,10]. To date, UG2UG research has been mostly conducted in laboratory ‘testbeds’ using repacked soils at prescribed soil moisture contents [7,11,12,13,14,15,16,17]. ‘Real world’ evaluation of WUSNs in outdoor environments are rare [8,9,10,18,19,20], and are even more limited in agricultural soils [5,21]. The lack of development of WUSNs compared to aboveground WSNs is largely due to the inherent difficulties associated with sending data through soil, and the need for extremely highly energy-efficient buried WUSN nodes, which cannot be recharged without excavation [5,6,7,22]. Soil is a difficult environment for radio frequency (RF) communication, as temperature, weather, soil moisture, soil composition, rocks, and foreign bodies such as pipes and building foundations all directly impact on communication performance [6,7,21]. Difficulty also arises from the highly heterogeneous nature of soils in which structure, texture, and volumetric soil moisture (VSM) differ over short intervals of location, depth, and time. For example [23] showed that signal attenuation through 50 cm of soil was 20 times greater than through air. The propagation of RF waves through a medium is inversely related to the dielectric constant ε_r_ (permittivity) of the medium, in which the dielectric constant of air is ε_r_ = 1, soil minerals are ε_r_ = 2–5 depending on minerology, and water is ε_r_ ≈ 80 [6,7,24,25]. 

The development of low-power wide-area network (LPWAN) technology, including LoRa, has presented new opportunities for the development of WUSNs. LoRa is designed for infrequent sending of small amounts of data (0.29 kbps to 50 kbps) over long distances (10+ km in rural areas) [6,26]. LoRa utilizes a chirp spread spectrum (CSS) modulation in which chirp frequency increases or decreases over time, allowing the signal to be demodulated, even when the received signal is 20 dB lower than the environmental noise floor [26]. In the last three years, a number of preliminary studies have explored the capability of 433-MHz LoRa for WUSN development [9,10,17,19,21,22,27]. However, as far as the authors are aware, 433-MHz LoRa for UG2AG or UG2UG applications have not been comprehensively investigated in ‘real world’ agricultural soils, nor has extensive evaluation of the effects of different soil types, data rates, power levels, or antennas been conducted.

## 2. Materials and Methods

### 2.1. Hardware

Hardware development was limited to ‘off the shelf’ components. The transmitter (TX) node consisted of an Adafruit Feather M0 RFM96 LoRa Radio – 433 MHz, with an ATSAMD21G18 processor, clocked at 48 MHz; 3.3 V logic; serial I2C, and an integrated 433-MHz RFM96 Semtech SX127x LoRa radio module. The buried receiver (RX) node consisted of an Adafruit Feather M0 RFM96 LoRa 433-MHz Radio; an Adafruit Adalogger FeatherWing data logger with a microSD socket and PCF8523 real-time clock; a 25.8 mm OLED display, and an Adafruit Ultimate GPS FeatherWing to allow the spatial tagging of logged data. Additionally, a remote-control (RC) node was developed using an Adafruit Feather M0 RFM96 LoRa 433-MHz Radio and 1/14 λ antenna for triggering the commencement of the TX transmission. Two antennae were evaluated: (i) a DexMRtiC 50 mm long, omnidirectional, 1/14 λ antenna, with a 2–3 dBi gain, <2.0 standing wave ratio (SWR), and 50 Ω impedance, and (ii) a DexMRtiC 220 mm long, directional, 1/3 λ antenna, which had a 12 dBi gain, <1.5 SWR, and 50 Ω impedance (Figure 1).

### 2.2. Software

For the initial trials at Site 1a (Kingston beach, Hobart (Rudosol)), the radio module’s modem was configured to transmit at 5.5 kbps (125 kHz bandwidth, 4:5 code ratio, 128 chirps/symbol spread factor and cyclic redundancy check (CRC) enabled) and 0.2 kbps (125-kHz bandwidth, a 4:8 code ratio, and a 4096 chirps/symbol spread factor). Data was transmitted at +5 dBm and +23 dBm power, with 40 packets per TX power level. The RX node used the last packet from the first half of the TX schedule as a synchronization trigger to reconfigure the RX node to the same data rate as the TX. Following the initial trials at Site 1a, all transmissions were conducted at 0.2 kbps (unless otherwise stated) at four TX power levels (+5, +10, +15, and +23 dBm), in which 10 data packets were sent per power level, resulting in 40 packets per TX-to-RX distance interval. The data logger recorded a timestamp, the packet number, TX power, received signal strength index (RSSI), frequency error (FE), and signal-to-noise ratio (SNR) (Figure 2). 

### 2.3. Field Sites

Trials were conducted between March 2018 and January 2019 in southern Tasmania on four contrasting soil types (Table 1). Initial evaluation of the software and hardware was conducted at Site 1a, Kingston Beach, Tasmania, (–42.9830°, 147.3242°) in beach sand (Rudosol). Sands are an ideal soil environment for radio wave transmission due to their low density, low clay content, and low moisture content. Later trials at the Kingston Beach site (Site 1b) in January 2019 explored the effect of soil moisture and salinity on transmission distance and quality by conducting transmissions approximately 30 m, 15 m, and 2 m away from the water’s edge. Site 2 was located at the University of Tasmania Farm, Cambridge, Tasmania (–42.7926, 147.4275), in a Brown Chromosol. The soil consisted of a sandy clay loam topsoil (0 to 23 cm), which abruptly overlaid an orange mottled medium clay subsoil. Radio transmission in Chromosols was expected to be problematic due to the contrasting moisture, clay content, and bulk density of the two upper soil layers. At Site 2, the effect of receiver height was explored by mounting the RX node on a 3 m telescopic pole. Site 3 was located at Saltwater River, Tasmania (–43.0211, 147.7210) in a Brown Dermosol, consisting of a brown clay loam to a depth of 28 cm, which had a gradual boundary to a yellow-brown medium clay with orange mottling and a heavy clay below 40 cm. Signal transmission at Site 3 was expected to be better than at Site 2 due to the porous and more homogeneous nature of Dermosols. However, at Site 3, measurements were not able to be obtained beyond 100 m from the TX node due to the spatial constraints of the site. Site 4 was located near Campbell Town, Tasmania (–41.9029, 147.4901) in a Red Ferrosol, consisting of a red clay loam to a depth of around 10 cm, which had a gradual change to a light sticky red clay subsoil. Radio transmission at Site 4 was expected to be impeded by the high free iron content (>6%) of the soil. Details of field sites are provided in Table 1. As far as the authors were able to establish, all sites were free of potential sources of electromagnetic interference, including buried water mains, fences, and existing RF devices.

### 2.4. Soil Properties

Dry bulk density (BD) was measured using a 6 cm × 6 cm core at 15 cm and 30 cm depth to correspond with the TX burial depth. Gravimetric soil moisture was determined by oven-drying at 105 °C for 24 h, and volumetric moisture content (VSM) was determined by multiplication with measured BD. Soil salinity was measured as a 1:5 soil water dilution as electrical conductivity (EC) using a Eutech Cond 6+ conductivity meter. Particle size was measured as the proportion of sand, silt (20–200 μm), and clay, by CSBP Limited Soil and Plant Analysis Laboratory, Western Australia using the sieve and pipette method.

### 2.5. Underground-to-Underground (UG2UG), and Underground-to-Aboveground (UG2AG) Treatments 

Field trials were conducted in two modes: (i) underground-to-aboveground (UG2AG) mode in which the TX node was buried at either 15 cm, 30 cm, or 50 cm depth, and the receiver node was positioned at 1.6 m above the ground surface (1.6 m and 3.0 m height at Site 2), at a range of distances up to 200 m away from the TX node (Table 2, Figure 3); and (ii) underground-to-underground (UG2UG) mode, in which the TX and RX nodes were buried at the same depth at either 15 cm, 30 cm, or 50 cm depth, at a range of transmission distances up to 50 m apart (Table 3, Figure 3). 

The extent to which UG2UG was able to be investigated was limited by the time required to excavate and rebury the RX node at each transmission distance. All UG2UG trials were conducted at 0.2 kbps data rate at 125 kHz bandwidth, a 4:8 code ratio, and 4096 chirps/symbol spread factor, at TX power levels of +5, +10, +15, +23 dBm, using a 1/14 λ TX antenna in the vertical orientation buried at 30 cm depth (Table 3). At Site 1b, the effect of TX and RX burial depth was also explored at 15 cm depth. At Site 4, TX burial depth was investigated at 50 cm and 85 cm depth, and the effects of TX–RX antenna orientation were also investigated (Table 3).

### 2.6. Data Analysis

Received signal quality was assessed by the received signal strength index (RSSI), frequency error (FE), and the signal-to-noise ratio (SNR). RSSI is a measure of the power present in a received radio signal, in which the higher (less negative or closer to zero) the RSSI value, the stronger the signal. For wireless networks, there is no agreed RSSI value at which networks are considered unusable, as RSSI is dependent on RX hardware sensitivity, RF protocols, and background noise. FE is the difference in frequency in Hz between the set operating frequency of the radio module and the received frequency, after adjustment for the effect of modulation and phase error. FE largely reflects the dielectric properties of the soil, in which variations result from phase change, fading, channel noise, and temperature variations between the TX and RX nodes. SNR is defined as the ratio of signal power to the noise power measured in dB or dBm. Positive values indicate more signal than noise.

Analysis of the degree to which different soil properties (soil type, moisture, EC_1:5_, sand %, etc.) and operational variables (TX depth, data rate, TX power, antenna type, etc.) influenced transmission quality, and distance maximum was complicated by the unbalanced nature of the trial design, missing data (different transmission distances between sites), and different data types, including binary (antenna type), integer (TX power, transmission distance, burial depth), and scaled variables (RSSI, SNR, soil moisture). In addition, all soil parameters were highly correlated. Consequently, multi-site analysis of the effects of soil and operational parameters were analyzed by Pearson correlation, rather than multivariate ANOVA. At specific sites and transmission distances, the effects of TX power and TX burial depth on RSSI, FE, and SNR were explored using one-way ANOVA, whilst the effects of binary variables (data rate, antenna type, and TX burial depth at 15 cm and 30 cm) were tested using an independent student T-test with a Bonferroni post hoc test in IBM SPSS^®^ version 25. Significant differences were identified when P < 0.05.

## 3. Results

### 3.1. Influence of Soil Properties and Operating Variables on UG2AG Communication

The received signal strength index (RSSI) at the maximum transmission distance at which packets were received was significantly correlated with soil particle size distribution (clay %, sand %, silt %) and volumetric soil moisture (VSM), but not any operational variables. Frequency error (FE) was highly correlated with all soil variables at all distances. The highest correlations occurred with VSM, clay content, and electrical conductivity (EC), as these properties are known to influence the dielectric constant. Of the operational parameters, FE was most closely correlated with data rate. The SNR was inconsistently correlated with a number of operational variables, although results were not consistent at multiple transmission distances (Table 4).

### 3.2. Effect of Soil Type and Soil Variables

Soil type had a profound effect on transmission distance (Figure 4). The maximum effective transmission distance was greatest for the Rudosol (Site 1b) at 200 m, followed by 170 m for the Chromosol (Site 2), and lastly 100 m for both the Dermosol (Site 3) and the Ferrosol (Site 4). However, it is noted that at Site 3 transmissions were not able to be conducted beyond 100 m, such that the ‘true’ maximum transmission distance could be determined. Soil type had an inconsistent effect on signal quality. At both 50 m and 100 m transmission distances, the RSSI for the Rudosol was significantly higher than for other soils, excluding the Dermosol. The unexpected variance in FE, RSSI, and signal-to-noise ratio (SNR) at Site 1 was attributed to an unknown source of RF interference between 0–50 m.

The effects of soil properties on transmission distance and quality were further explored at Site 1b, as particle size (sand, silt, and clay %) was consistent, whilst VSM and EC varied with distance from the water’s edge. EC and VSM were significantly correlated (R^2^ = 0.882), and thus comparisons between soil factors were confounded. Figure 5 demonstrates that signal quality (RSSI, FE, and SNR) was more strongly associated with VSM (R^2^ = 0.826 to 0.539) than with EC (R^2^ = 0.535 to 0.319), in which increasing VSM from 9.2% to 30.9% (and increasing EC from 0.51 dS/m to 4.06 dS/m) reduced RSSI by 42 dBm, reduced SNR by 21, and increased FE by 452 Hz. This increase in VSM and EC also decreased the maximum distance at which data was received from 200 m to 30 m (data not shown).

### 3.3. Effect of Data Rate on Transmission Quality and Distance

The 0.2 kbps data rate resulted in data packets being received at a maximum of 200 m when the TX node was buried at 15 cm depth, and 100 m when the TX node was buried at 30 cm depth, compared to only 150 m and 50 m respectively at the 5.5 kbps data rate. At Site 1b (Rudosol), the effect of data rate on RSSI and SNR was highly variable. For example, at +23 dBm power and 15 cm TX depth, RSSI was significantly lower than at 5.5 kbps at distances of 10 m, 20 m, and 150 m, yet significantly higher at 50 m, whilst having no significant effect at 100 m (Figure 6).

### 3.4. Effect of TX Burial Depth on Transmission Quality and Distance

Increasing the depth of TX burial from 15 cm to 30 cm significantly decreased RSSI and SNR and significantly increased FE (0.2 kbps data rate) at all transmission distances, and all TX power levels, at all four sites (except Site 4, at +23 dBm power at 20 m and 30 m). TX burial depth significantly affected the maximum transmission distance at Sites 1b, 2, and 4 (at Site 3, it was not possible to test reception beyond 100 m). The maximum transmission distance when the TX node was buried at 15 cm depth at Sites 1b, 2, and 4 were 200 m, 170 m, and 100 m, respectively, compared to 100 m, 100 m, and 70 m when the TX node was buried at 30 cm, and 10 m when the TX node was buried at 50 cm depth at Site 1b (Figure 7).

### 3.5. Effect of TX Power on Transmission Quality and Distance

TX power significantly affected RSSI and the maximum distance at which data was able to be received. At 30 cm TX depth (and 0.2 kbps data rate), the maximum distance that a signal was able to be received at 5 dBm was 30 m at Site 4, and 50 m at Sites 1, 2, and 3. In contrast, at 23 dBm, the maximum distance that a signal was able to be received was 100 m at Sites 1b and 2, 75 m at Site 3, and 70 m at Site 4. At 30 cm TX depth, the RSSI was significantly higher at 23 dBm than at 5 dBm and 15 dBm at all transmission distances, and at all sites (Figure 8).

One-way ANOVA found that a sequential increase in TX power (i.e., +5 to +10 dBm, +10 to +15 dBm, etc.) significantly increased RSSI in 193 out of 219 combinations of site (4), power level (4), transmission distance (≤9), and TX depth (3). Of the 26 non-significant combinations, 21 occurred at Site 1b where RF interference was suspected. Results for SNR were similar to those of RSSI: one-way ANOVA demonstrated that a sequential increase in TX power significantly increased SNR in 190 out of 219 combinations, with exceptions largely occurring at Site 1b. 

### 3.6. Effect of RX Antenna Type and RX Height on Transmission Quality and Distance

At Site 1b, use of the lower specified 1/14 λ RX antenna significantly reduced the maximum transmission distance from 100 m (–113 dBm, SNR –2.2) to 50 m (–112 dBm, SNR –1.0) at TX depth 15 cm, (5.5 kbps data rate, +23 dBm TX power). Differences in RSSI, FE, and SNR at all transmission distances were significant. At Site 3 (Dermosol), the effect of tRX antenna on transmission distance was compared at +5 dBm TX power (at 0.2 kbps data rate), in which use of the lower specified 1/14 λ RX antenna significantly reduced the maximum transmission distance from ≥100 m (RSSI –131 dBm, SNR –17) to 70 m (–135 dBm, SNR –18) at a TX depth of 15 cm, and from 50 m (RSSI –132 dBm, SNR –16) to 15 m (–133 dBm, SNR –19) at a TX depth of 30 cm. RX antenna type significantly affected RSSI and SNR at all TX power levels, at all TX burial depths, and at all transmission distances (Figure 9).

The influence of TX antenna on signal quality and distance only received rudimentary exploration during the initial trial at Site 1a. At a 5.5 kbps data rate and 5 dBm TX power, the larger 1/3 λ TX antenna allowed the reception of intact data packets at a maximum distance of 30 m (–120 dBm RSSI), whilst the smaller 1/14 λ TX antenna supported the reception of intact packets at a maximum distance of 10 m (–131 dBm RSSI).

The effect of RX antenna height on signal quality was explored at Site 2 (Chromosol) (TX depth 30 cm, 0.2 kbps data rate). When the RX node was mounted at a 1.6 m height, data was only received at 70 m at +15 or +23 TX power, yet when the RX node was mounted at a 3.0 m height, data was received between 70–100 m at all four power levels. At both + 15 and +23 dBm TX power, increased RX height significantly increased the RSSI and SNR at 70 m, 90 m, and 100 m, and FE at 100 m.

### 3.7. Comparison between UG2UG and UG2AG Operation on Transmission Quality and Distance

Transmission distance and signal quality (RSSI and SNR) were significantly shorter and poorer for UG2UG transmissions compared to UG2AG transmissions at all sites, at all TX power levels, and at all transmission distances (Figure 10 versus Figure 4). The maximum distance at which an UG2UG data packet was received (0.2 kbps data rate, +23 dBm TX power, and 30 cm RX and TX depth) occurred at Site 1b (Rudosol) at 20 m, followed by Site 4 (Ferrosol) at 8 m, and Site 3 (Dermosol) at 5 m (Figure 10), compared to 100 m, 45 m, and 70 m, respectively, for UG2AG transmissions (Figure 8). At Site 1b when the RX and TX nodes were buried at 15 cm depth, the maximum distance at which data packages were received was 50 m (data not shown), compared to 200 m for UG2AG transmission. 

UG2UG transmission distance and signal quality were greatly affected by soil type and soil properties. RSSI, FE, and SNR were significantly correlated with all soil and operating variables. The highest correlation with RSSI existed with VSM at R^2^ = –0.591, P<0.001, the highest correlation with FE existed with a node burial depth at R^2^ = 0.615, P < 0.001 followed by BD at R^2^ 0.509, P < 0.001, whilst the highest correlation with SNR existed with VSM at R^2^ = –0.719, P < 0.001. 

Data from all sites indicated that UG2UG signal transmission improved at higher TX power levels. For example, at Site 3 (Dermosol), increasing the TX power from +5 dBm to +23 dBm increased the transmission distance from 1.0 m to 5.0 m, and increased RSSI from –119 to –104 at 1 m. Whilst at Site 4 (Ferrosol), no data was received at or beyond a 2.0 m transmission distance when the TX power was +5 dBm, yet data was received at 6 m when sent at +23 dBm. UG2UG data transmission was also greatly affected by TX and RX burial depth. At Site 1b, TX and RX burial at 15 cm depth resulted in a signal being received at 50 m, whilst burial at 30 cm depth resulted in data being received at a maximum of 20 m distance. Likewise, at Site 4 (Ferrosol), data was received at 8 m when the TX and RX nodes were buried at 30 cm depth, but was not received at 2 m when the RX node was buried at 50 cm and 85 cm depths. Data from Site 4 also indicates that UG2UG data transmission was slightly improved (increased maximum transmission distance by 2 m, RSSI by 7 dBm, and SNR 0.3) when the TX and RX antennas were horizontally oriented toward each other (data not shown). 

## 4. Discussion

Results for our LoRa 433-MHz band underground-to-underground (UG2UG) communication were disappointing. At 30 cm burial depth, the maximum transmission distances were 20 m at Site 1b, 5 m at Site 3, and 8 m at Site 4. These results are similar to previous findings such as [8], who reported that the UG2UG 433 MHz MoleNet was only able to reliably communicate over 7.5 m, and [6] who concluded that practical UG2UG link distances are still limited to around 12 m. 

The maximum underground-to-aboveground (UG2AG) transmission distance when the transmitter (TX) node was buried at 15 cm depth ranged between 200 m at Site 1b to 100 m at Site 4, and between 70 m at Site 4 to 100 m at Sites 1, 2 and 3 when the TX node was buried at 30 cm depth. These distances are similar to those of other studies. Using 433-MHz LoRa, [10] reported a maximum UG2AG transmission distance between 250–300 m at a 10% packet survival rate, whilst [8] reported that the 433 MHz MoleNet system was able to communicate up to 80 m when the TX node was buried at 20 cm depth, and [21] found that the LoRa Wizard was able to communicate over 300 m. Differences in the maximum transmission distance between Site 1 and Site 4 were not unexpected, as transmission at Site 1 was through beach sand (1.9% clay) at very low volumetric moisture content (9.2% VSM), compared to moist (43% VSM) light clay and clay loam (26.0% clay) at Site 4. The highest signal quality (signal-to-noise ratio (SNR) and received signal strength index (RSSI)) occurred at Site 3 (Dermosol), despite this site having the highest clay content, highest VSM, and second-highest electrical conductivity (EC). This finding is in contrast to previous studies that have demonstrated that increases in these soil attributes, especially VSM, negatively impact on RF transmission quality and distance [6,7,11,16]. Whilst it is attractive to attribute the high transmission quality at Site 3 to the site having the lowest bulk density (BD), the correlation analysis indicated BD to be one of the least related soil properties to both UG2UG and UG2AG signal quality. The poorer than expected signal quality at Site 1b was attributed to its semi-urban location, which appeared to be subject to more radio-frequency interference than the other three rurally located sites. 

Attributing the maximum UG2AG transmission distance and signal quality to soil and operational parameters was difficult due to the unbalanced nature of the trial design, different variable types, and high level of correlation between variables. Results of the correlation analysis cautiously indicated that UG2UG communication quality was significantly influenced by VSM and the burial depth of the TX and receiver (RX) nodes. For UG2AG communication, RSSI and SNR were more strongly influenced by operational parameters, specifically TX depth and TX power, compared to soil parameters. In contrast, frequency error (FE) was more strongly related to soil properties, especially VSM. At the maximum received distance, RSSI and SNR were significantly correlated with VSM; however, VSM was not significantly correlated with RSSI and SNR at any discrete transmission distance. At Site 1b, in which the distance of the TX node from the water’s edge was used to vary VSM and EC whilst keeping the particle size and BD constant, UG2AG results demonstrated that RSSI, SNR, and FE were highly related to VSM and, to a lesser extent, soil salinity. Increasing VSM from 9.2% to 30.9% and EC from 0.51 dS/m to 4.06 dS/m decreased the maximum transmission distance from 200 m to 30 m, decreased the RSSI by 40 dBm, decreased the SNR by 19, and increased the FE by 440 Hz. This finding is supported by a number of laboratory testbed studies that have concluded that soil moisture is the most important soil property affecting UG2UG transmission quality and maximum distance at 433 MHz [7,14,16]. For example, [17] found that increasing soil moisture from 5% to 22% reduced RSSI by 13 dBm, [7] found increasing VSM from 5% to 30% increased path loss by 28 dB, and [14] found that increasing VSM from 5% to 30% reduced RSSI by 43 dBm. Reduced signal quality at higher moisture contents results from the diffusion, attenuation, and adsorption of radio frequency waves [6]. Increased VSM also increases soil permittivity, which shortens the signal’s wavelength during travel through soil; thus, at a given transmission frequency, the wavelength is not constant [16]. 

Of the different operational parameters, burial depth was found to have the greatest effect on UG2AG transmission distance and quality. Increasing burial depth from 15 cm to 30 cm decreased the maximum UG2AG transmission distance by between 30% to 50% at all sites (100 m to 30 m), and decreased the average RSSI by 18 dBm and the average SNR by 12. The effect of burial depth on UG2AG transmission quality and distance is complex, as increasing the burial depth increases the signal attenuation due to the greater transmission path length from the TX node to the soil surface, and because increased soil depth is often associated with increased VSM, EC, clay content, and BD. At Site 1b (0.2 kbps, +23 dBm), increasing the burial depth from 15 cm to 30 cm and then to 50 cm reduced the maximum transmission distance from 200 m to 100 m to 10 m, whilst VSM increased from 9.2% to 13.6% to 28.2%. At Site 4, increasing the TX burial depth from 15 cm to 30 cm decreased the maximum transmission distance from 100 m to 70 m and reduced the RSSI at 70 m by 12.8 dBm, yet VSM remained unchanged. In contrast to our findings, a small number of UG2AG studies have reported that increasing the TX burial depth had only a minor effect on transmission quality and distance. For example, [18] showed that increasing the TX burial depth from 5 cm to 20 cm had less than a 5 dBm effect on the RSSI, whilst [8] reported that increasing the TX burial depth from 15 cm to 55 cm reduced the RSSI by only 10 dBm. UG2UG studies also demonstrate that transmission distance and quality have a complex and variable response to burial depth. Researchers demonstrated that RSSI initially decreased, and then increased before decreasing again as the node depth increased from the soil surface to 100 cm depth, resulting in an overall reduction in RSSI of around 45 dBm [14]. In contrast, [15] demonstrated that signal strength decreased linearly with depth between 10–100 cm, resulting in RSSI decreasing by 20 dBm. Notably, most UG2AG and UG2UG studies on the effect of burial depth on signal quality have either been conducted in uniform repacked soils, which do not vary like field soils. 

Increasing TX power from +5 dBm to +23 dBm significantly increased the maximum transmission distance by between 30–150 m (TX buried at 15 cm depth), and improved the signal quality at all sites by an average of 18 dBm RSSI, 90 Hz FE, and 12 SNR. The effect of TX power on UG2AG communication has not previously been reported. The effect of TX power on UG2UG signal quality was investigated by [7], who reported that increasing the transmit power from +5 to +10 dBm increased the RSSI by around 3 dBm in dry soil, and around 10 dBm in moist soil, depending on the internode distance (TX and RX nodes at 30 cm depth).

The effect of data rate on signal quality was only investigated at Site 1b. Decreasing the data rate from 5.5 kbps to 0.2 kbps increased the transmission distance by 50 m, yet had a highly variable effect on RSSI and SNR, whilst significantly reducing FE. This may have been due to differences in packet air time between the two data rates, resulting in differences in the measurement of RSSI and SNR. Studies by [21] and [10] indicate that increasing the packet size from 8 bytes to 64 bytes decreased the LoRa propagation distance by around 30 m, and decreased the packet success rate by around 0% to 20%, depending on the burial depth and VSM, which were both shown to be more important than packet size.

The results demonstrated that for UG2AG transmissions, the use of the higher specification 1/3 λ antenna at the TX node had less effect on transmission distance and quality than using the higher specified antenna at the RX node. Use of the 1/3 λ RX antenna at Sites 1a and 3 increased the maximum distance at which data packets were received by 100 m and 30 m, respectively. In all cases, the 1/3 λ RX antenna significantly improved the RSSI and SNR. The ‘real world’ study by [5] also demonstrated that the use of circular, planar, and Yagi (10 dB) antennae could improve the UG2AG communication range from 14 m to 65 m, compared to more conventional single-ended elliptical and full-wave dipole antennae. 

Mounting the RX node at 3.0 m as opposed to 1.6 m increased the transmission distance by 50 m to 100 m (+5 and +10 dBm, 0.2 kbps), yet had little effect on the RSSI and SNR. In UG2UG transmissions, antenna orientation improved the maximum transmission distance by only 2 m and had a minimal effect on signal quality. This result differs from that of [7], who found that the UG2UG packet error rate differed between 10–100% depending on the RX antenna angle, and [12], who reported that the antenna orientation increased the ultrawide-band UG2AG signal attenuation at 7 GHz by up to 40 dB. The use of antennae designed for transmission through air are not necessarily suited for use in soil, as the burial of antennae changes their characteristics due to the high dielectric constant of the soil, resulting in variations in resonant frequency, which is shifted to a lower spectrum, and system bandwidth [6]. 

## 5. Conclusions

Results of this study demonstrate that for nodes buried at 30 cm depth, underground-to-underground (UG2UG) data communication at distances between 5–20 m are possible using 433-MHz LoRa in relatively dry soil conditions. These distances do not represent a substantial improvement upon previous studies. At these distances, 433-MHz LoRa-based UG2UG applications in agriculture appear largely impractical, as ideal hop distances need to be in the order of hundreds of meters to have practical applications.

This study demonstrated that in relatively dry soil conditions, 433-MHz LoRa WUSNs were able to communicate underground-to-aboveground (UG2AG) up to a distance between 100–200 m, depending on soil type, volumetric soil moisture (VSM), and operational variables. Signal quality (received signal strength index (RSSI) and signal-to-noise ratio (SNR)) was mostly influenced by the node-to-node transmission distance, transmitter (TX) burial depth, TX power, data rate, and receiver (RX) antenna type. To a lesser extent, signal quality was influenced by soil parameters including: VSM, electrical conductivity (EC), and soil particle size. In this study, UG2AG communication distance and quality were maximized at a TX depth of 15 cm and at a TX power of +23 dBm; using a 0.2 kbps data rate and a 1/3 λ RX antenna mounted 3.0 m above the ground in a relatively dry (VSM 9.0%), low clay content (1.9%) beach sand (Rudosol).

The results presented in this manuscript suggest that existing 433-MHz LoRa technology is highly suitable for the development of UG2AG WUSNs for agricultural applications. However, further development is likely to be constrained by the energy efficiency and battery life of the TX node. Results from this study suggest that gains in signal quality (SNR and RSSI) and transmission distance can be achieved through increasing the TX power, reducing the data rate, and the use of higher specification RX antennae. Given that increasing the transmission power from +5 dBm to +23 dBm resulted in a 62.5× increase in power consumption, and decreasing the data rate from 5.5 kbps to 0.2 kbps resulted in a 27.5× increase in power, the best approach for improving communication quality and distance is likely to be through the use of the highest possible specification RX antenna, followed by the optimization of the data rate. Fortunately for most agricultural applications, increasing the size and power demands of the RX node are unlikely to be problematic. Further research is required to determine the effect of weather and seasonal variations in VSM on signal quality and transmission distance. Additionally, research avenues to minimize power consumption at the TX node and optimize below-ground TX antenna design remain open.

## Figures and Tables

**Figure 1 sensors-19-04232-f001:**
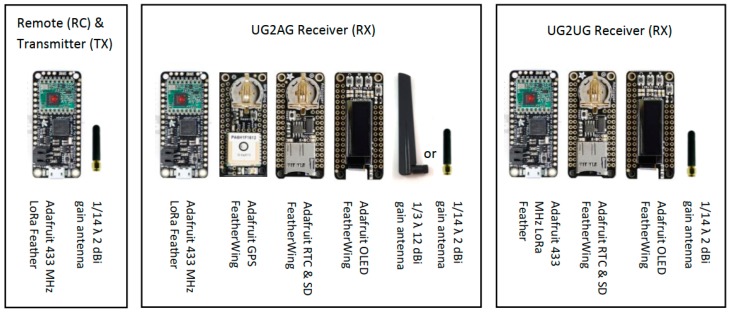
Hardware components for underground-to-aboveground (UG2AG) and underground-to-underground (UG2UG) nodes.

**Figure 2 sensors-19-04232-f002:**
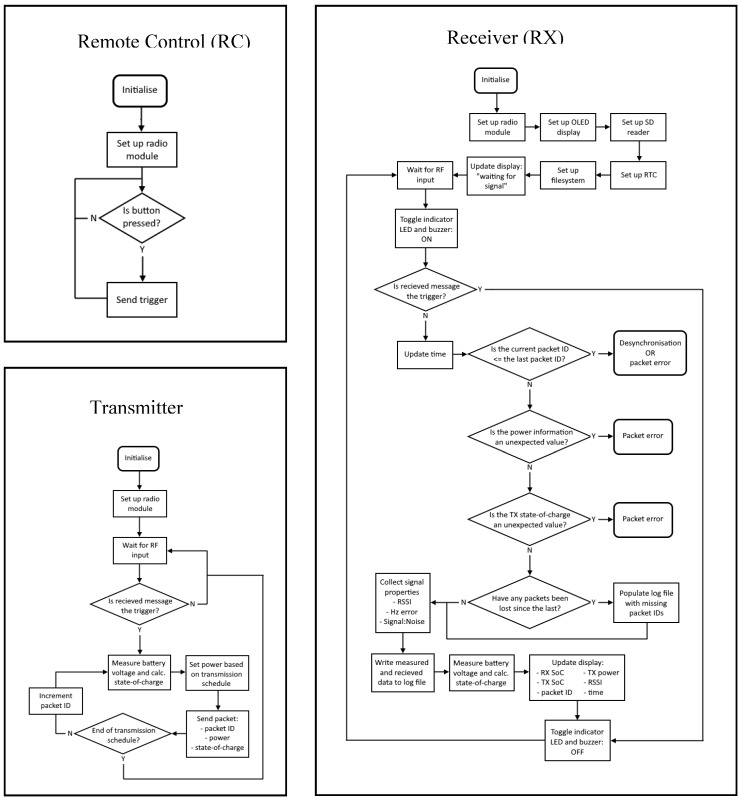
Flow diagram of remote control, transmitter and receiver node software.

**Figure 3 sensors-19-04232-f003:**
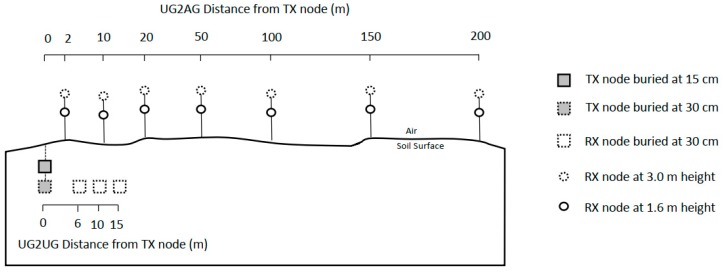
Schematic of field trial layout. Note distance between the TX and receiver (RX) nodes for both UG2AG and UG2UG experiments varied between sites. RX measurement at 3.0 m was only conducted at Site 2.

**Figure 4 sensors-19-04232-f004:**
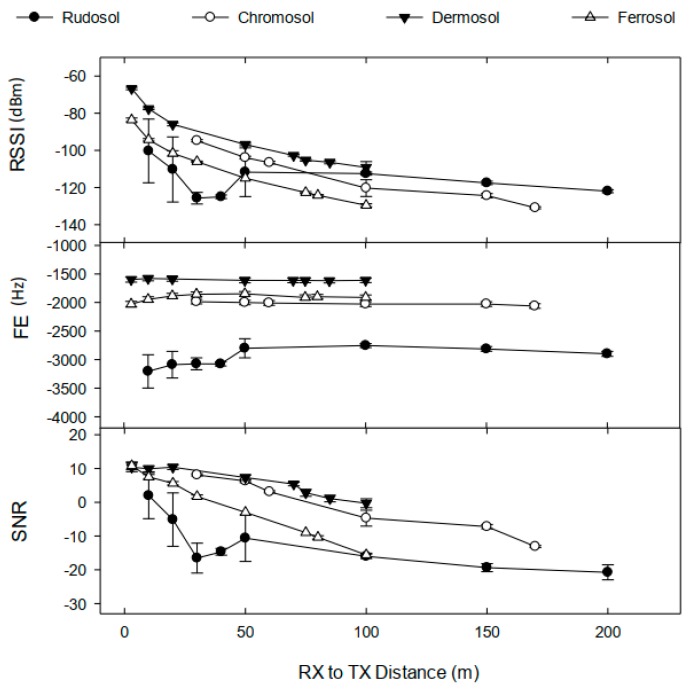
Effect of soil type on RSSI, FE, and SNR vs. RX to TX distance. Note: Values represent the average received RSSI, FE and SNR (+23 dBm TX power, TX buried at 15 cm depth, 0.2 kbps data rate, and 1/3 λ RX antenna). RSSI (dBm) refers to received signal strength index, FE (Hz) refers to frequency error, and SNR refers to signal-to-noise ratio. Error bars indicate ±1 standard deviation.

**Figure 5 sensors-19-04232-f005:**
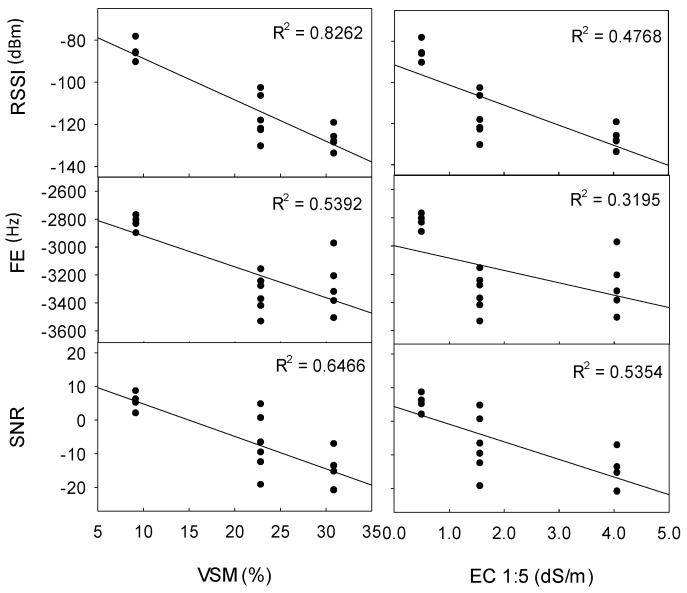
Effect of soil moisture (VSM) and salinity (EC 1:5) on signal quality at Site 1b (Rudosol) at transmission distances of 10 m, 20 m, and 30 m (TX power +15 dBm and +23 dBm, 0.2 kbps data rate, 1/14 λ TX antenna, 1/3 λ RX antenna). RSSI (dBm) refers to received signal strength index, FE (Hz) refers to frequency error, and SNR refers to signal-to-noise ratio. Error bars indicate ±1 standard deviation.

**Figure 6 sensors-19-04232-f006:**
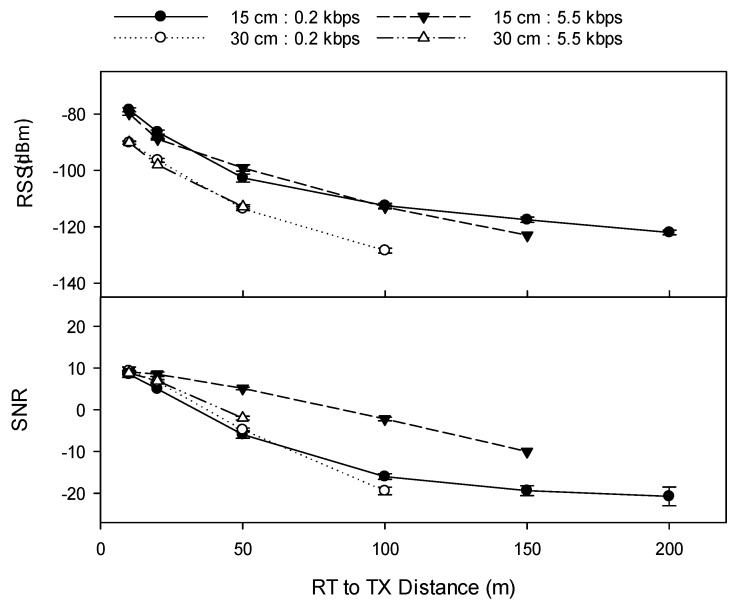
Effect of data rate (kbps), and TX burial depth (cm) on transmission distance and quality at Site 1b (Rudosol) (TX power +23 dBm, 1/14 λ TX antenna, 1/3 λ RX antenna). Error bars indicate ±1 standard deviation. RSSI (dBm) refers to received signal strength index, SNR refers to signal-to-noise ratio. Error bars indicate ±1 standard deviation.

**Figure 7 sensors-19-04232-f007:**
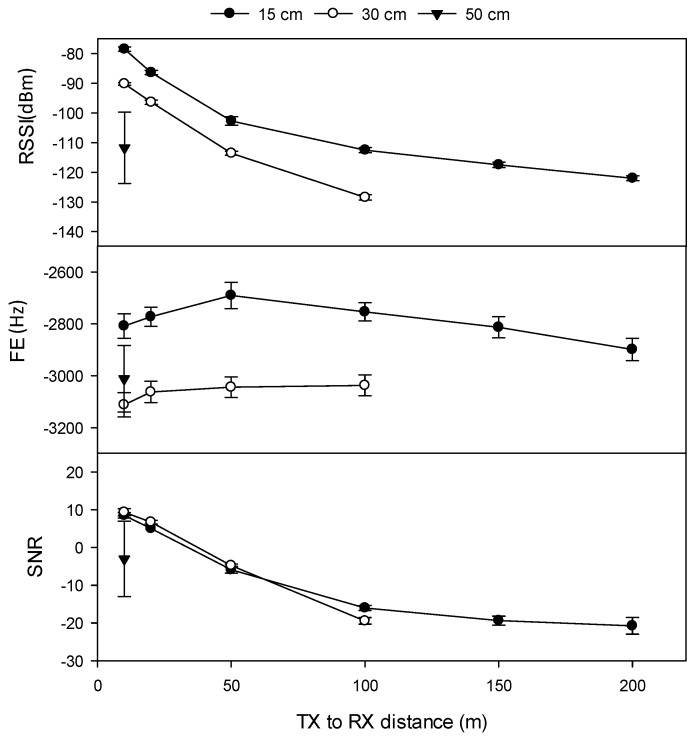
Effect of TX burial depth (cm) on transmission distance and quality at Site 1b (Rudosol) (0.2 kbps data rate, TX power +23 dBm, 1/14 λ TX antenna, 1/3 λ RX antenna). RSSI (dBm) refers to received signal strength index, FE (Hz) refers to frequency error, SNR refers to signal-to-noise ratio. Error bars indicate ±1 standard deviation.

**Figure 8 sensors-19-04232-f008:**
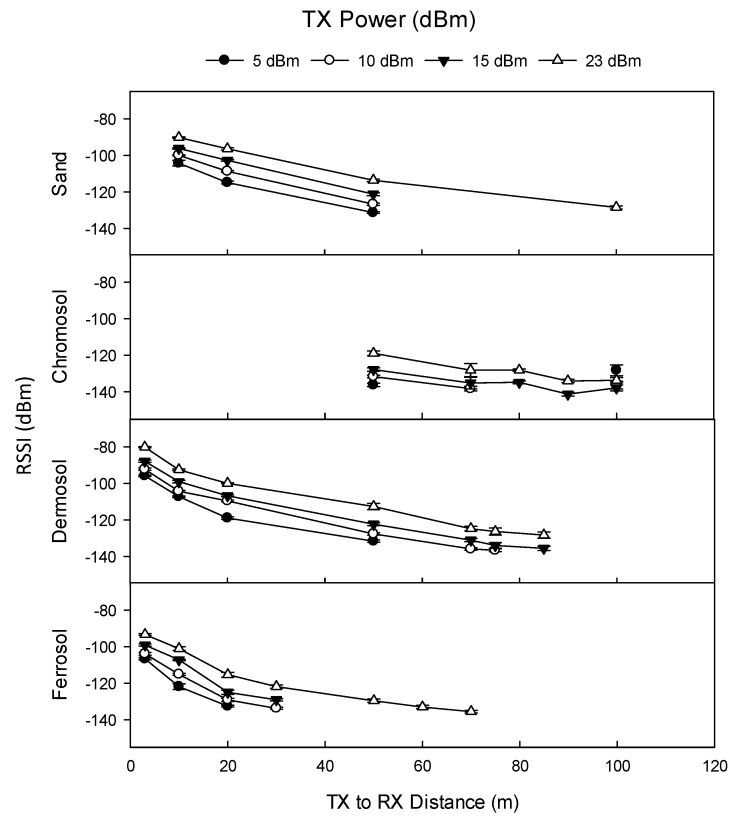
Effect of TX power (dBm) on RSSI by transmission distance by site (0.2 kbps data rate, 30 cm TX burial depth, 1/14 λ RX antenna). Sand is equivalent to Rudosol. RSSI (dBm) refers to received signal strength index, FE (Hz) refers to frequency error, and SNR refers to signal-to-noise ratio. Error bars indicate ±1 standard deviation.

**Figure 9 sensors-19-04232-f009:**
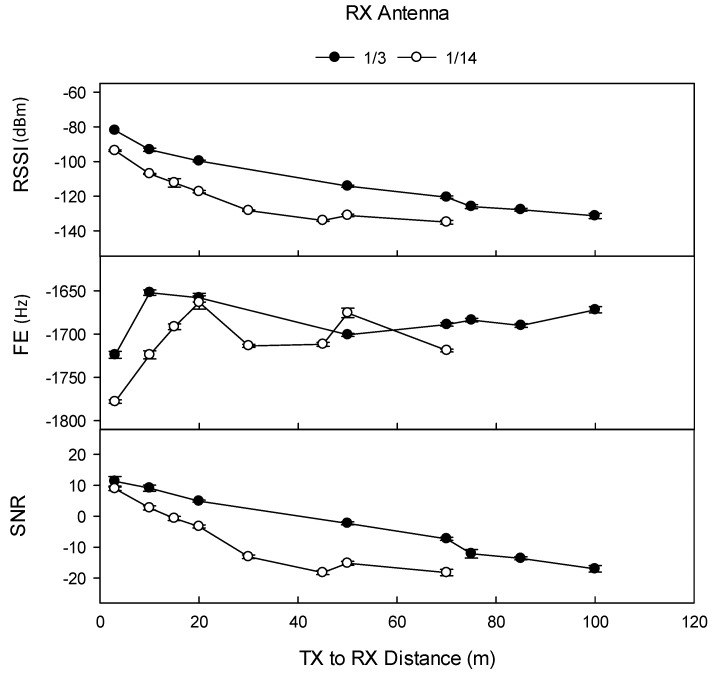
Effect of RX antenna type on signal quality, Site 3 (Dermosol) (0.2 kbps data rate, TX buried at 15 cm depth, +5 dBm TX power, TX antenna 1/14 λ, RX antenna 1/3 λ, and 1/14 λ). Note that the maximum investigated transmission distance was 100 m. RSSI (dBm) refers to received signal strength index, FE (Hz) refers to frequency error, and SNR refers to signal-to-noise ratio. Error bars indicate ±1 standard deviation.

**Figure 10 sensors-19-04232-f010:**
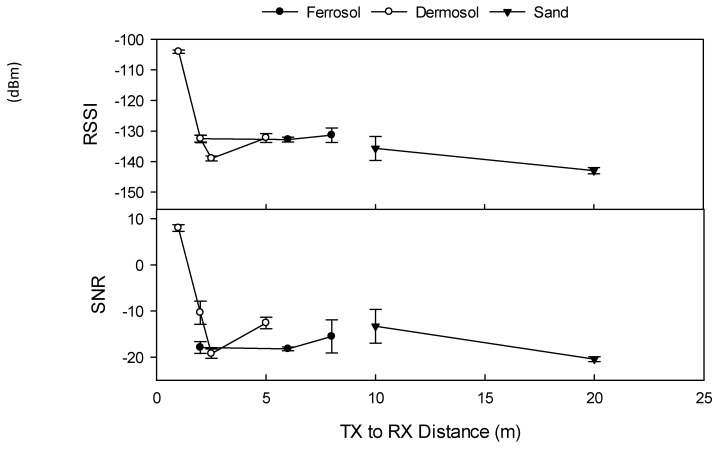
UG2UG transmission distance and quality (0.2 kbps data rate, RX and TX 1/14 λ antenna, +23 dBm TX power, TX and RX depth of 30 cm). RSSI (dBm) refers to received signal strength index, FE (Hz) refers to frequency error, and SNR refers to signal-to-noise ratio. Error bars indicate ±1 standard deviation.

**Table 1 sensors-19-04232-t001:** Field site characteristics.

Field Site	1 a&b	2	3	4
Location	Kingston Beach	Cambridge	Saltwater River	Campbell Town
Soil Type	Rudosol	Brown Chromosol	Brown Dermosol	Red Ferrosol
GPS	–42.9830°, 147.3242°	–42.7926°, 147.4275°	–43.0211°, 147.7210°	–41.9029°, 147.4901°
Land use	Recreation	Irrigated pasture	Dryland pasture	Irrigated crop/pasture
Trial date	(a) Mar 2018, (b) Jan 2019	Jul 2018	Aug 2018	Aug 2018
Site Characteristic	Ideal for transmission	Two contrasting soil layers	Uniform low-density soil	High iron content, poor transmission
Topsoil Texture	Sand	Sandy Clay Loam	Sandy Clay Loam	Clay Loam
Subsoil Texture	Sand	Medium Clay	Clay Loam	Light Clay
Max. Distance Tested	250 m	170 m	100 m	100 m
UG2AG Trial	Yes	Yes	Yes	Yes
UG2UG Trial	Yes	No	Yes	Yes
% Clay 15 cm	1.9	8.2	32.5	26.0
% Clay 30 cm	1.2	43.8	74.5	30.8
% Sand 15 cm	98.1	86.7	50.6	54.2
% Sand 30 cm	96.4	51.4	18.5	51.9
VSM (%) 15 cm	9	33	47	43
VSM (%) 30 cm	13	30	49	43
EC (dS/m) 15 cm	0.5	0.16	0.29	0.15
EC (dS/m) 30 cm	1.15	0.15	0.51	0.31
BD (g/cm^3^) 15 cm	1.53	1.37	0.89	1.16
BD (g/cm^3^) 30 cm	1.54	1.70	1.25	1.56

BD is bulk density, VSM is volumetric soil moisture, EC is electrical conductivity as a 1 to 5 soil water dilution. UG2AG is underground to above ground communication mode, UG2UG is underground to underground mode. Soil classification according to [28]. 15 and 30 cm refer to the depth of sampling and corresponding transmitter (TX) burial depth.

**Table 2 sensors-19-04232-t002:** Details of underground-to-aboveground (UG2AG) trials.

Site	Data Rate (kbps)	TX Burial Depth (cm)	TX Power (dBm)	TX Antenna λ	RX Antenna λ
1a	5.5	30	+23	1/14	1/3
1a	5.5	30	+5	1/14	1/3
1a	5.5	30	+5	1/3	1/3
1a	5.5	50	+23	1/14	1/3
1b	0.2	15	+5, +10, +15, +23	1/14	1/3
1b	5.5	15	+5, +10, +15, +23	1/14	1/3
1b	5.5	15	+5, +10, +15, +23	1/14	1/14
1b	0.2	30	+5, +10, +15, +23	1/14	1/3
1b	5.5	30	+5, +10, +15, +23	1/14	1/3
1b	0.2	50	+5, +10, +15, +23	1/14	1/3
1b	5.5	50	+5, +10, +15, +23	1/14	1/3
1c	0.2	15	+5, +10, +15, +23	1/14	1/3
2	0.2	15	+5, +10, +15, +23	1/14	1/3
2	0.2	30	+5, +10, +15, +23	1/14	1/3
3	0.2	15	+5, +10, +15, +23	1/14	1/3
3	0.2	15	+5, +10, +15, +23	1/14	1/14
3	0.2	30	+5, +10, +15, +23	1/14	1/3
3	0.2	30	+5, +10, +15, +23	1/14	1/14
4	0.2	15	+5, +10, +15, +23	1/14	1/3
4	0.2	30	+5, +10, +15, +23	1/14	1/3

Site 1 Kingston beach (Rudosol), Site 2 Cambridge (Chromosol), Site 3 Saltwater River (Dermosol), Site 4 (Ferrosol). Site 1a refers to initial trials in March 2018, 1b refers to later trials in January 2019, and 1c refers to trials located, 30 m, 15 m, and 2 m from the water’s edge.

**Table 3 sensors-19-04232-t003:** Details of underground-to-underground (UG2UG) trials.

Site	TX Burial Depth (cm)	TX Antenna λ	RX Burial Depth (m)	RX Antenna λ	RX Orientation	Transmission Distance (m)
1b Rudosol	15	1/14	15	1/14	Vertical	10, 20, 30, 40, 50
	30	1/14	30	1/14	Vertical	10, 20, 30
2 Chromosol	30	1/14	30	1/3	Vertical	3, 6, 10, 15
3 Dermosol	30	1/14	30	1/14	Vertical	5
	30	1/14	30	1/14	Vertical	2.5, 5
	30	1/14	30	1/14	Vertical	1, 2, 3, 4
4 Ferrosol	30	1/14	30	1/14	Horizontal	2, 4, 8, 10, 15
	30	1/14	30	1/14	Horizontal	4, 6, 8, 10, 15
	30	1/14	30	1/14	Vertical	2, 6
	30	1/14	50	1/14	Vertical	6
	30	1/14	85	1/14	Vertical	6

All trials were conducted using transmission power of +5, +10, +15, +23 dBm, and a 0.2 kbps data rate at a 125-kHz bandwidth, 4:8 code ratio, and 4096 chirps/symbol spread factor. Antenna orientation, horizontal refers to the RX and TX antennae being parallel to the ground, while vertical refers to the RX antenna being perpendicular to the ground.

**Table 4 sensors-19-04232-t004:** Pearson correlation (R values) of soil properties and operational parameters on transmission distance and signal quality.

		Received Signal Strength Index				Frequency Error				Signal to Noise Ratio			
		Max. Dist.	Transmission Distance			Max. Dist.	Transmission Distance			Max. Dist.	Transmission Distance		
			10 m	50 m	100 m		10 m	50 m	100 m		10 m	50 m	100 m
**Soil Properties**	Site					0.826	0.706	0.675	0.536				
	Sand %	0.350				−0.813	−0.725	−0.777	−0.644				
	Clay %	−0.354				0.786	0.731	0.785	0.637				
	Silt %	0.311				0.835	0.660	0.693	0.608				
	BD (g/cm^3^)					−0.530	−0.508	−0.453					
	VSM (%)	−0.495				0.764	0.840	0.844	0.770	−0.365			
	EC1:5		−0.391			−0.491	0.713	−0.869	−0.848		−0.463		
**Operational Variables**	TX depth		−0.562	−0.423	−0.484						−0.516	−0.308	
	TX power		0.396	0.571							0.319	0.552	
	RX antenna		−0.379								−0.285		
	Data rate					−0.358	−0.661	−0.654	−0.813	0.407			

Note: Only significant correlations are shown, P < 0.05. Data collected at 5.5 kbps data rate and initial trials at Site 1a were excluded from the analysis. ‘Site’ refers to site number as a covariate. VSM is volumetric soil moisture. EC 1:5 stands for electrical conductivity at 1:5 soil water dilution. Distance (m) refers to the distance between the TX and RX nodes as discrete analysis. Max. Dist. refers to the maximum distance at which a data package was received.

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
