# Peer review of "Underground Wireless Data Transmission Using 433-MHz LoRa for Agriculture"

_sensors, 2019, doi:10.3390/s19194232_

Round 1

Reviewer 1 Report

This article addresses an underground (UG) wireless data transmission, which is used the LoRa-based LP-WAN techniques.

Both UG2UG and UG2AG are evaluated in four in situ soils. The adopted hardware and its software are described in Section 2.1 and 2.2, but the information seems not to be enough. The authors should provide a comprehensive WSN system architecture. Some figures of the evaluated WSN system should be provided. How about sensors? What the UG data measured? It seems to be only shown the transmission distances test.

Misc. Page 1: "MEA" Plexus mesh.... should be "Measurement Engineering Australia (MEA) Plexus mesh....", moreover, a reference should be provided.

Author Response

The authors should provide a comprehensive WSN system architecture. Some figures of the evaluated WSN system should be provided.

The reviewer’s concerns have been addressed. Three new figures have been added to improve explanation of the field site procedure and testbed architecture, including demonstrating hardware components and explaining software.

How about sensors? What the UG data measured? It seems to be only shown the transmission distances test.

The testbed was not connected to any sensors, as the research was focused only on data communication performance. Radio packet information was dynamic, however, as described in section 2.2 and Figure 2. Addition of sensors is considered a trivial addition to future research.

Misc. Page 1: "MEA" Plexus mesh.... should be "Measurement Engineering Australia (MEA) Plexus mesh....", moreover, a reference should be provided.

Defined acronym. Additional details on MEA and Terralytic sensors are provided in text.

Reviewer 2 Report

In this paper, a real-world verification of underground wireless data transmissions is conducted with 433MHz LoRa for agriculture.Four field sites are selected as the testing environments. While the paper provides the real-world data which is quite promising, there are still some issues that should be addressed in detail. 

1. The presentation needs to be improved. For example, there should be a summary paragraph in the first section to overview the remaining sections. Some abbreviations are used without mentioning the whole terminologies, e.g., VSM, CSBP laboratories. The result figures should be reformatted to ease the readability.

2. In page 3, the paper mentioned the initial trials at Site 1a, but there is no detail for this testing site. Perhaps it's a typo.

3. Authors should study the real-world data with existing analysis/testbed results. The corresponding discussion will help readers understand the relationship between the modeling/emulation and the real-world testing.

4. Authors could consider more sophisticated designs of underground communications to optimize the transmission qualities for UG2UG scenarios. For example, the paper below gave a cross-layer communication protocol designs to significantly boost the data rates in underground environments.

"Distributed Cross-Layer Protocol Design for Magnetic Induction Communication in Wireless Underground Sensor Networks," IEEE Transactions on Wireless Communications, vol. 14, no. 7, pp. 4006-4019, July 2015. 

Author Response

The presentation needs to be improved. For example, there should be a summary paragraph in the first section to overview the remaining sections.

With respect, the authors consider the abstract provides sufficient summary of the manuscript. An additional summary would only further add to the length of the manuscript. We request the issue be referred to the editors. 

Some abbreviations are used without mentioning the whole terminologies, e.g., VSM, CSBP laboratories..

Acronyms defined for each major section throughout the text. CSBP Limited is the actual trading name of the company. Added full name: CSBP Limited Soil and Plant Analysis Laboratory.

The result figures should be reformatted to ease the readability

How should the result figures be reformatted? The authors are unsure how to make the figures any simpler. Added missing units in axis labels.

In page 3, the paper mentioned the initial trials at Site 1a, but there is no detail for this testing site. Perhaps it's a typo.

The text has been amended to “…Kingston beach, Hobart (Rudosol) (Site 1a).” Details of the sites are described in detail in section 2.3.

Authors should study the real-world data with existing analysis/testbed results.

The authors respectfully suggest that this request is in error as the shortcomings of testbed studies vs real world studies are discussed in the introduction Ln 45-55, and our results are compared to the following testbed studies:

Vuran, M. C.; Silva, A. R., Communication Through Soil in Wireless Underground Sensor Networks – Theory and Practice. In Sensor Networks. Signals and Communication Technology, Springer, Berlin, Heidelberg, 2009; pp 309–347.

Salam, A.; Vuran, M. C., Impacts of soil type and moisture on the capacity of multi-carrier modulation in internet of underground things. 2016 25th International Conference on Computer Communications and Networks, ICCCN 2016.

Zemmour, H.; Baudoin, G.; Diet, A., Soil effects on the underground-to-aboveground communication link in ultrawideband wireless underground sensor networks. IEEE Antennas and Wireless Propagation Letters 2017, 16, 218-221.

Yu, X.; Wu, P.; Zhang, Z.; Wang, N.; Han, W., Electromagnetic wave propagation in soil for wireless underground sensor networks. Progress In Electromagnetics Research M 2013, 30, 11-23.

Yu, X. Q.; Zhang, Z. L.; Han, W. T. Evaluation of communication in wireless underground sensor networks, In IOP Conference Series: Earth and Environmental Science, 2017.

Dong, X.; Vuran, M. C. Impacts of soil moisture on cognitive radio underground networks, In 2013 1st International Black Sea Conference on Communications and Networking, BlackSeaCom 2013, 2013; pp 222-227.

The corresponding discussion will help readers understand the relationship between the modeling/emulation and the real-world testing.

Whilst the authors acknowledge the value of modelling/emulation studies, development of their own was considered outside the scope of the present study.  Results from the present study could be applied to future modelling efforts.

Authors could consider more sophisticated designs of underground communications to optimize the transmission qualities for UG2UG scenarios. For example, the paper below gave a cross-layer communication protocol designs to significantly boost the data rates in underground environments. "Distributed Cross-Layer Protocol Design for Magnetic Induction Communication in Wireless Underground Sensor Networks,"IEEE Transactions on Wireless Communications, vol. 14, no. 7, pp. 4006-4019, July 2015. 

We were not able to obtain the full paper within the 10 days required for resubmission of the manuscript. However, from the abstract, citations of the paper and figures within the paper, we consider the topic to be beyond the scope of the present study. Our study was based on LoRa 433 MHz communication, whilst the paper in question is focused on magnetic induction. Also, the UG2UG testbed described only extended 15 meters which is not a great advancement on existing RF approaches. The paper demonstrates how a large sensor network utilising mesh-network behaviour can save energy, which is great, but our experience with aboveground meshed networks was not encouraging. In agriculture, we are not interested in meshed solutions. As such, we are only considering single-hop typologies. Furthermore, data rate is trivial for agricultural WUSNs as the systems we are seeking to develop are likely to send only very small amounts of data (e.g. soil moisture) only a few times a day; anything more is superfluous.

Regarding cross-layer protocols, our study was entirely focused on the physical interaction between RF generated and received by LoRa 433MHz components and in situ soils.  If we had developed a fully-fledged network with sensors, we could have designed network protocols that responded dynamically to the network environment, as others have before.

Reviewer 3 Report

 The authors of this paper present the development and results of a 433 MHz low-range (LoRa) wireless sensor network to assess underground to underground (UG2UG) as well as to underground to aboveground (UG2AG) wireless communications. Four types of soils have been studied in order to analyze important wireless communications metrics. The results report interesting findings according to the soil type. This is one of the first contributions in this research direction. 

The paper is well structured and flows well. Moreover, it is well written. Only a few details should be corrected regarding the paper's presentation. For example, lines 36 and 55 contain sentences starting with a reference, which could be much better written. In line 81, "a Adafruit" should be corrected to "an Adafruit ...". Minimal details like these must be corrected in order to provide a better reading.

Regarding the contribution, the results are quite interesting. Section 2, "Materials and Methods" starts by describing the hardware and software of the proposal. Therefore, in order to present a much more solid contribution, both sections (2.1 and 2.2) must be developed in more detail. The hardware section should at least provide a basic diagram of the modules comprising the system. The same for the software section. Besides, even if at the beginning of the paper the authors specify that different types of soils are studied, in Section 2.2 the description immediately starts by labeling with "Site 1a" the first type of soil. Thus, please consider to provide first a description of each type of soil and then feel free to refer to them.

Author Response

lines 36 and 55 contain sentences starting with a reference, which could be much better written.

Fixed.  No sentences begin with references.

 In line 81, "a Adafruit" should be corrected to "an Adafruit ...". Minimal details like these must be corrected in order to provide a better reading.

The whole manuscript has been re-edited to address spelling and grammatical errors.

Sections (2.1 and 2.2) software and hardware must be developed in more detail. The hardware section should at least provide a basic diagram of the modules comprising the system.

A diagram of the modules used is now included in the manuscript.

The same for the software section.

A diagram of the software design is now included in the manuscript.

At the beginning of the paper the authors specify that different types of soils are studied, in Section 2.2 the description immediately starts by labeling with "Site 1a" the first type of soil. Thus, please consider to provide first a description of each type of soil and then feel free to refer to them.

These corrections have been made

Round 2

Reviewer 1 Report

The authors have great efforts for enhancing and improving their first submitted draft. The authors have satisfyingly replied to my all questions. So, I think that the revised manuscript can be accepted for publication.

Misc.

The quality of Figure 2 should be improved.